# Potential factors influencing COVID-19 vaccine acceptance and hesitancy: A systematic review

**Debendra Nath Roy** [1☯¤], **Mohitosh Biswas** [2‡], **Ekramul Islam** [2‡], **Md. Shah Azam** [3,4☯] *

**1** Department of Pharmacy, Jashore University of Science and Technology, Jashore, Bangladesh,
**2** Department of Pharmacy, University of Rajshahi, Rajshahi, Bangladesh, **3** Department of Marketing,
University of Rajshahi, Rajshahi, Bangladesh, **4** Vice Chancellor, Rabindra University, Sirajganj, Bangladsh

☯ These authors contributed equally to this work.
¤ Current address: Institute of Education and Research, University of Rajshahi, Rajshahi, Bangladesh
‡ These authors also contributed equally to this work.
* mdshah.azam@yahoo.com.au

**Data Availability Statement:** All relevant data are within the paper and its Supporting Information files.

**Funding:** No external fund was available

## Abstract

### Background and aims

Although vaccines are considered the most effective and fundamental therapeutic tools for consistently preventing the COVID-19 disease, worldwide vaccine hesitancy has become a widespread public health issue for successful immunization. The aim of this review was to identify an up-to-date and concise assessment of potential factors influencing COVID-19 vaccine acceptance and refusal intention, and to outline the key message in order to organize these factors according to country count.

### Methods

A systematic search of the peer-reviewed literature articles indexed in reputable databases, mainly Pub Med (MEDLINE), Elsevier, Science Direct, and Scopus, was performed between 21st June 2021 and 10th July 2021. After obtaining the results via careful screening using a PRISMA flow diagram, 47 peer-reviewed articles met the inclusion criteria and formed the basic structure of the review.

### Results

In total, 11 potential factors were identified, of which the greatest number of articles (n = 28) reported "safety" (34.46%; 95% CI 25.05—43.87) as the overarching consideration, while "side effects" (38.73%; 95% CI 28.14—49.32) was reported by 22 articles, which was the next common factor. Other potential factors such as "effectiveness" were identified in 19 articles (29.98%; 95% CI 17.09—41.67), followed by "trust" (n = 15 studies; 27.91%; 95% CI 17.1—38.73), "information sufficiency" (n = 12; 34.46%; 95% CI 35.87—63.07), "efficacy" (n = 8; 28.73%; 95% CI 9.72—47.74), "conspiracy beliefs" (n = 8; 14.30%; 95% CI 7.97—20.63), "social influence" (n = 6; 42.11%; 95% CI 14.01—70.21), "political roles" (n = 4; 16.75%; 95% CI 5.34—28.16), "vaccine mandated" (n = 4; 51.20%; 95% CI 20.25—82.15),

**Competing interests:** There is no competing interests to declare.

and "fear and anxiety" (n = 3; 8.73%; 95% CI 0.59—18.05). The findings for country-specific influential vaccination factors revealed that, "safety" was recognized mostly (n = 14) in Asian continents (32.45%; 95% CI 19.60—45.31), followed by the United States (n = 6; 33.33%; 95% CI12.68—53.98). "Side effects" was identified from studies in Asia and Europe (n = 6; 35.78%; 95% CI 16.79—54.77 and 16.93%; 95% CI 4.70—28.08, respectively), followed by Africa (n = 4; 74.60%, 95% CI 58.08—91.11); however, public response to "effectiveness" was found in the greatest (n = 7) number of studies in Asian countries (44.84%; 95% CI 25—64.68), followed by the United States (n = 6; 16.68%, 95% CI 8.47—24.89). In Europe, "trust" (n = 5) appeared as a critical predictor (24.94%; 95% CI 2.32—47.56). "Information sufficiency" was identified mostly (n = 4) in articles from the United States (51.53%; 95% CI = 14.12—88.74), followed by Asia (n = 3; 40%; 95% CI 27.01—52.99). More concerns was observed relating to "efficacy" and "conspiracy beliefs" in Asian countries (n = 3; 27.03%; 95% CI 10.35—43.71 and 18.55%; 95% CI 8.67—28.43, respectively). The impact of "social influence" on making a rapid vaccination decision was high in Europe (n = 3; 23.85%, 95% CI -18.48—66.18), followed by the United States (n = 2; 74.85%). Finally, "political roles" and "vaccine-mandated" were important concerns in the United States.

## Conclusions

The prevailing factors responsible for COVID-19 vaccine acceptance and hesitancy varied globally; however, the global COVID-19 vaccine acceptance relies on several common factors related to psychological and, societal aspect, and the vaccine itself. People would connect with informative and effective messaging that clarifies the safety, side effects, and effectiveness of prospective COVID-19 vaccines, which would foster vaccine confidence and encourage people to be vaccinated willingly.

## Introduction

The corona virus disease 2019 (COVID-19) has been an unprecedented disease burden around the world that has drastically impacted diverse areas of human societies, from public health systems to, education, economic growth, and personal well-being. As of the end of the first week of August 2021, more than 200 million confirmed cases and more than 4.2 million deaths caused by the disease have been reported worldwide [1]. Public health authorities are searching for preventive strategies to limit the spread of corona viruses because an effective treatment for the COVID-19 disease is not yet to be available, [2–4]. Since the pandemic poses a significant disease burden to health systems and a threat to the global health, along with preventive community measures, massive immunization is considered the most powerful and cost-effective health intervention, as well as the most promising strategy to combat this contagious virus and to save human lives. According to the Centers for Disease Control and Prevention (CDC), to date, vaccines are the most powerful therapeutic tools available to curb the spread of infectious viruses such as COVID-19 [5]; however, promoting effective vaccine candidates and achieving public acceptance are urgent matter and public health priorities that must be satisfied to successfully manage COVID-19.

After the new corona virus emerged in 2019, using past experiences many scientists around the world focused their endless efforts into quickly developing an effective vaccine.

Impressively, since last year an unprecedented number of 74 vaccine candidates have been developed, which have successfully passed through clinical trials and are included in COVID-19 vaccine platform. The World Health Organization (WHO) and Food and Drug Administration (FDA) have approved 3 candidates to date, and granted conditional approval for 7 more candidates in phase three trials [6, 7]. As such, alongside an implementable and equitable vaccine distribution policy, ensuring the vaccine acceptance of a new vaccine by the general public is equally important, because it has been reported that, the real uptake rate of a pandemic vaccine could be much lower than the expected values [8, 9]. For example, in the H1N1 influenza pandemic, the acceptance rates of a newly lunched vaccine were seen to range from 17 to 67%, even in many developed countries [8–10].

Although vaccination has been one of the most important interventions in the field of public health throughout the 21st century, worldwide COVID-19 vaccine hesitancy is a prevalent issue and is viewed as one of the top ten global public health challenges [11]. Vaccine hesitancy refers to the reluctance or unwillingness to get vaccinated or unwillingness to administer vaccines to one's children against an infectious disease, even if the vaccine is proven to be safe and, effective and the service is assessable to uptake the vaccine [12]. Vaccine hesitancy is expressed in "3C" sequences, which point to confidence, complacency, and convenience. The World Health Organization Strategic Advisory Group of Experts (WHO-SAGE) defines vaccine hesitancy as a "delay in acceptance or refusal of vaccines, despite the availability of vaccination services" [13]. Actually, low initial vaccine uptake intention to a particular vaccine or vaccination program is a psychological and dynamic phenomenon observed through global perspectives [14]. The extent to which and how clearly an individual understand the relevance of the pandemic vaccine significantly depends on trust, which in turn is related to personal beliefs, motivation, perceived risk exposure, knowledge, and awareness of the vaccination [14]. A highly effective vaccine was found to have strong acceptance [15], while vaccine with low effectiveness could negatively impact on uptake intention and reduce the willingness to receive the new vaccine [16]. Resource less and marginalized peoples and disadvantaged minority group have previously been less likely to be vaccinated for influenza [17]. During a crude vaccine optimization process, inadequate vaccine safety data diminished the vaccine confidence index and produced distrust in health services, public health experts and state agencies. Moreover, widespread fake news on vaccines and the vaccination process, misinformation, and propaganda were identified as several key determinants of global vaccine refusal [18]. Taken together, an effective intervention is needed to improve public acceptance and trust of COVID-19 vaccines, to ease concerns over the safety, side effects, and benefits of vaccines; and target inoculation campaigns in disadvantaged and marginalized groups who have already been seriously affected by COVID-19 [19]. In this regard, frequent communication between health workers and remote population groups is also important to address the hesitancy-associated predictors and to motivate vaccine-hesitant individuals towards vaccine acceptance [20].

The current evidence confirming that, best-practice community interventions, such as the use of face masks, good hand hygiene, and maintaining social distancing, are effective ways of preventing the rapid spread of COVID-19 in low-and middle-income countries (LMICs) [21]; however, optimization of crude immunization through an effective vaccine is the ultimate therapeutic tool in useful public health interventions against the COVID-19 disease [5]. Public willingness to accept a newly promoted vaccines varies with space, social class, time, ethnicity and contextual human behavior as reported in previous studies [14, 22, 23]; therefore, in order to implement a vaccine-based community health intervention nationwide, the primary aim is to understand the common factors that lead to COVID-19 vaccine hesitancy and refusal

intention globally, because a lag in the vaccination process in LMICs could facilitate the spread of new variants of COVID-19 to rest of the world.

To date, however, most of the systematic reviews and meta-analysis is performed on COVID-19 vaccination have focused on the assessment of vaccine acceptance or rejection rates [19, 24, 25] and few studies have tried to summarize the factors that most influence COVID-19 vaccine acceptance intention and refusal among the different countries. As a result, there is a paucity of systematic reviews describing the most common factors influencing COVID-19 vaccine uptake or refusal intention, with the factors varying by country count globally; hence, this systemic review aimed to identify and highlight the most common factors of COVID-19 vaccine uptake and refusal intention and to, summarize the key drivers that influence the complex motives behind COVID-19 vaccine hesitancy among individuals in different continents.

## Materials and methods

In this review the relevant factors and themes associated with the COVID-19 vaccine acceptance or hesitancy concerns were examined. We searched scholarly peer-reviewed databases to identify and design a framework of the probable factors influencing hesitancy to uptake a new vaccine aimed at COVID-19 infection. The screening procedure involved a flow diagram in accordance with the Preferred Reporting Items for Systematic Reviews and Meta Analyses- (PRISMA) 2020 [26] statement for new systematic reviews of databases and for the literature selection process. The inclusion criteria were the following: 1) peer-reviewed published articles from electronic databases including Pub Med (MEDLINE), Elsevier, Embase, Science Direct, Scopus and other reputed resources; 2) survey studies involving all types of sample populations; (3) the scope and principal aim of the study was to identify the potential factors influencing COVID-19 vaccine acceptance and hesitancy; (4) publication studies in the English language. The exclusion criteria were the following: (1) unpublished manuscripts; (2) the article did not publish the required data related to vaccine acceptance and refusal factors; (3) the publication language was not in English. To understand the complex interplay of a wide variety of intervening factors for COVID-19 vaccine acceptance and hesitancy, this study was aimed to identify potential factors influencing COVID-19 vaccine acceptance and hesitancy across the world. The search items we used in this study were adopted from recently published articles on COVID-19 vaccine acceptance and hesitancy, and systematic review focused on the assessment of COVID-19 vaccine acceptance and rejection rate. The literature search for peer-reviewed articles was conducted on21st June 2021 to 10th July by using the keywords: "COVID-19 vaccine hesitancy" OR"COVID-19 vaccine hesitancy and associated factors" OR "COVID-19 vaccine confidence"OR "COVID-19 vaccine AND acceptance intention" using a descriptive style. In addition, the references for the studies that met the inclusion criteria were searched to include additional articles in the review. The initial searches of peer reviewed records from electronic databases produced 98 articles, 5 of which articles were identified from reference lists from included articles. All authors independently assessed the documents during the inclusion process. In the initial phase and before screening, 6 duplicate articles and 1 review article were recognized and removed from the process while 96 articles were screened. During the eligibility assessment, 11 articles were removed after the abstract screening step. After full text assessment, 38 articles were excluded by the independent reviewer due to lacking key searched data set that meet the study objectives; hence 47articles were selected for the review and final analysis to explore the potential factors associated with COVID-19 vaccine uptake intention and hesitancy. The types of papers included in the study were mostly cross-sectional survey research papers. The search strategy used for potential factor identification

involved identifying multiple key factors from individual articles, collecting the respondent's number of each variable, and calculating the respondent's mean (%) against the total mean value. From the sample, standard deviation (SD) and standard error (SE) were calculated to show the 95% confidence interval (CI).

## Results

"Fig 1" shows the PRISMA statement flow diagram for the literature search and selection process. After removal of duplicates, the independent reviewer abstract screening process resulted in 93.20% of initial agreement on which abstracts were satisfactory for the purpose of the study. After the application of inclusion criteria during the abstract screening and full-text assessment of the eligibility stages, finally 48.96% studies meeting initial agreement criteria were included in the final analysis.

In this review, the sample populations we analyzed were from different countries in Asia (Bangladesh, India, China, Jordan, Saudi Arabia, Qatar, Israel, Kuwait, Turkey), Europe (Portugal, Slovenia, Poland, Germany, France, United Kingdom), the United States, and Africa (Uganda, Zambia, Middle East, Egypt, involving multi-ethnic (report of 19 and 22 countries) backgrounds and LMICs (9 countries and Asia, Africa, and South America). The characteristics of the study participants included general populations, industrial service workers, self-employed workers, university employees, service personnel, farmers, managers and administrators, associate professionals, clerical support workers, service and sales workers, craft and related workers, plant and machine operators and assemblers, elementary occupations, private

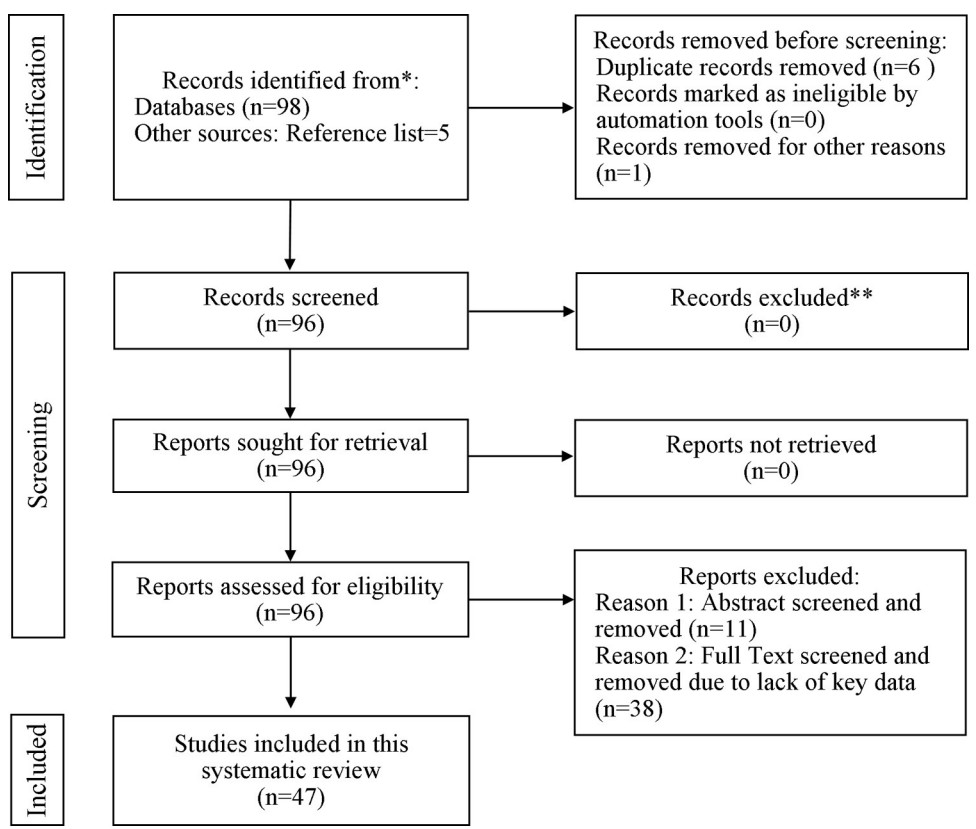

**Fig 1. PRISMA-based flow diagram of study selection process for new systematic reviews.**

workers, government workers, monthly paid job holders, agricultural employees, business people, day-laborers, house wives, unemployed people, health professionals, students in various backgrounds, adolescents, young adults, older adults, and various ethnicities.

The most frequently identified key factors in COVID-19 vaccine acceptance and refusal are illustrated in "Table 1". Since we identified multiple factors from each individual article in

**Table 1. Potential factors associated with COVID-19 vaccine acceptance and hesitancy.**

| Factors | Authors [Count] | Mean total populations ($\bar{X}$) | Mean respondents ($\bar{x}$), (95% CI) | Mean respondents ($\bar{x}$%) (95% CI) |
|---|---|---|---|---|
| Safety | Soares et al., 2021 [27]; Jain et al., 2021 [28]; Lin et al., 2020 [29]; Wang K et al., 2021[30]; Suresh et al., 2021 [31]; Abedin et al., 2021 [32]; Bai et al., 2021 [33]; El-Elimat et al., 2021 [34]; Cai et al., 2021 [35]; Almaghaslah et al., 2021 [36]; Silva et al., 2021 [37]; Manning et al., 2021 [38]; Sharun et al., 2020 [39]; Palm et al., 2021 [40]; Pogue et al., 2020 [41]; Wang J et al., 2020 [42]; Al-Mulla et al., 2021 [43]; Kanyike et al., 2021 [44]; Petravić et al., 2021 [45]; Grochowska et al., 2021 [46]; Rosental&Shmueli, 2021 [47]; Jiang et al., 2021 [48]; Mudenda et al., 2021 [49]; Lazarus et al., 2021 [50]; Faezi et al., 2021 [51]; Nikolovski et al., 2021 [52]; Burhamah et al., 2021 [53]; Holzmann-Littig et al., 2021 [54] | 2088.64 | 513.29 (275.87—750.72) | 34.46 (25.05—43.87) |
| Efficacy | Jain et al., 2021 [28]; Lin et al., 2020 [29]; Almaghaslah et al., 2021 [36]; Kanyike et al., 2021 [44]; Nikolovski et al., 2021 [52]; Tavolacci et al., 2021 [55]; Kose et al., 2021 [56]; Freeman et al., 2021 [57] | 2851.75 | 556.38 (-2.55—1115.31) | 28.73 (9.72—47.74) |
| Information sufficiency | Soares et al., 2021 [27]; Lin et al., 2020 [29]; Suresh et al., 2021 [31]; Almaghaslah et al., 2021 [36]; Silva et al., 2021 [37]; Kanyike et al., 2021 [44]; Nikolovski et al., 2021 [52]; Sherman et al., 2021 [58]; Saied et al., 2021 [59]; Riad et al., 2021 [60]; Kaplan et al., 2021 [61]; Lucia et al., 2020 [62] | 2246.33 | 1333.67 (397.18—2270.16) | 49.47 (35.87—63.07) |
| Trust | Soares et al., 2021 [27]; Jain et al., 2021 [28]; El-Elimat et al., 2021 [34]; Kanyike et al., 2021 [44]; Petravić et al., 2021 [45]; Riad et al., 2021 [60]; Lazarus et al., 2021 [50]; Kose et al., 2021 [56]; Freeman et al., 2021 [57]; Holzmann-Littig et al., 2021 [54]; Lucia et al., 2020 [62]; Mascarenhas et al., 2021 [63]; Kelekar et al., 2021 [64]; Grüner&Krüger, 2020 [65]; Padhi& Al-Mohaithef, 2021 [66] | 2678.80 | 635 (103.37—1166.63) | 27.91 (17.1—38.73) |
| Side effect | Suresh et al., 2021 [27]; Bai et al., 2021 [33]; El-Elimat et al., 2021 [34]; Manning et al., 2021 [38]; Kanyike et al., 2021 [44]; Petravić et al., 2021 [45]; Rosental&Shmueli, 2021 [47]; Jiang et al., 2021 [48]; Mudenda et al., 2021 [49]; Faezi et al., 2021 [51]; Nikolovski et al., 2021 [52]; Holzmann-Littig et al., 2021 [54]; Tavolacci et al., 2021 [55]; Kose et al., 2021 [56]; Freeman et al., 2021 [57]; Sherman et al., 2021 [58]; Saied et al., 2021 [59]; Riad et al., 2021 [60]; Lucia et al., 2020 [62]; Bono et al., 2021 [67]; Szmyd et al., 2021 [68]; Arce et al., 2021 [69] | 3303.55 | 1209.36 (516.85—1901.87 | 38.73 (28.14—49.32) |
| Effectiveness | Wang K et al., 2021 [30]; Abedin et al., 2021 [32]; El-Elimat et al., 2021 [34]; Almaghaslah et al., 2021 [36]; Silva et al., 2021 [37]; Sharun et al., 2020 [39]; Palm et al., 2021 [40]; Pogue et al., 2020 [41]; Wang J et al., 2020 [42]; Al-Mulla et al., 2021 [43]; Grochowska et al., 2021 [46]; Mudenda et al., 2021 [49]; Lazarus et al., 2021 [50]; Nikolovski et al., 2021 [52]; Holzmann-Littig et al., 2021 [54]; Saied et al., 2021 [59]; Lucia et al., 2020 [62]; Bono et al., 2021 [67]; Reiter et al., 2021 [70] | 2817.32 | 508.74 (243.31—774.18) | 29.38 (17.09—41.67) |
| Conspiracy beliefs | Pogue et al., 2020 [41]; Lazarus et al., 2021 [50]; Burhamah et al., 2021 [53]; Szmyd et al., 2021 [68]; Islam et al., 2021 [71]; Sallam et al., 2021a [72]; Sallam et al., 2021b [73] | 2843.13 | 426.38 (-18.43—871.19) | 14.30 (7.97—20.63) |
| Social influence | Lin et al., 2020 [29]; Cai et al., 2021 [35]; Holzmann-Littig et al., 2021 [54]; Tavolacci et al., 2021 [55]; Freeman et al., 2021 [57]; Mascarenhas et al., 2021 [63] | 2924.83 | 1172.50 (-73.52—2418.52) | 42.11 (14.01—70.21) |
| Political roles | Palm et al., 2021 [40]; Holzmann-Littig et al., 2021 [54]; Riad et al., 2021 [60]; Reiter et al., 2021 [70] | 3567 | 521.25 (102.55—939.95) | 16.75 (5.34—28.16) |
| Vaccine-mandate | Almaghaslah et al., 2021 [36]; Silva et al., 2021[37]; Lucia et al., 2020 [62]; Mascarenhas et al., 2021 [63] | 378.50 | 194 (37.94—350.06) | 51.20 (20.25—82.15) |
| Fear & anxiety | Rosental&Shmueli, 2021 [47]; Nikolovski et al., 2021 [52]; Holzmann-Littig et al., 2021 [54]; | 4176.67 | 180 (60.17—299.82) | 8.73 (-0.59—18.05) |

response to COVID-19 vaccine acceptance intention and hesitancy, in total 11 potential factors were identified from 47 articles [27–73], among which the most articles (n = 28) reported "safety"(respondent's mean ($\bar{x}$) = 513.19; 95% CI 275.87─750.72, respondent's mean ($\bar{x}\%$) = 34.46; 95% CI 25.05─43.87; total sample populations mean ($\bar{X}$) = 2088.64) as the overarching concern, while "side effects" ($\bar{x}$ = 1209.36, 95% CI 516.85─1901.87; $\bar{x}\%$ = 38.73, 95% CI 28.14─49.32, and $\bar{X}$ = 3303.55) was identified in 22 studies as influencing COVID-19 vaccination intention. Of the other key factors, "effectiveness" was identified in 19 articles ($\bar{x}$ = 508.74, 95% CI 243.31─774.18; $\bar{x}\%$ = 29.98, 95% CI 17.09─41.67and $\bar{X}$ = 2817.32); followed by "trust" (n = 15; $\bar{x}$ = 635, 95% CI 103.37─1166.63; $\bar{x}\%$ = 27.91, 95% CI 17.1─38.73 and $\bar{X}$ = 2678.80); "information sufficiency" (n = 12; $\bar{x}$ = 1333.67, 95% CI 397.18─2270.16; $\bar{x}\%$ = 34.46, 95%, CI 35.87─ 63.07 and $\bar{X}$ = 2246.33), while 8 articles reported both "efficacy" ($\bar{x}$ = 556.38, 95% CI -2.55─1115.31; $\bar{x}\%$ = 28.73, 95% CI 9.72─47.74 and $\bar{X}$ = 2851.75) and "conspiracy beliefs"($\bar{x}$ = 426.38, 95% CI -18.43─871.19; $\bar{x}\%$ = 14.30, 95% CI7.97─20.63 and $\bar{X}$ = 2843.13). "social influence" (n = 6) was another key factor ($\bar{x}$ = 1172.5, 95% CI -73.52─2418.52; $\bar{x}\%$ = 42.11, 95% CI 14.01─70.21 and $\bar{X}$ = 2924.83). The terms "political roles" and "vaccine-mandated" were reported by 4 studies ($\bar{x}$ = 521.25, 95% 102.55─939.95; $\bar{x}\%$ (%) = 16.75, 95% CI5.34─28.16 and $\bar{X}$ = 3567; $\bar{x}$ = 194, 95% CI 37.94─350.06; $\bar{x}\%$ = 51.20, 95% CI 20.25─82.15, $\bar{X}$ = 378.50 respectively) respectively. Finally "fear and anxiety" was also identified as a potential factor (n = 3; $\bar{x}$ = 180, 95% CI 60.17─299.82; $\bar{x}\%$ = 8.73, 95%, CI 0.59─18.05 and $\bar{X}$ = 4176.67) as shown in Table 1.

"Table 2" summarizes and describes the mode of distribution frequency of these key factors around the world. Following Table 1, the results revealed that "safety" was recognized mostly (n = 14) in Asian countries ($\bar{x}$ = 496.93, 95% CI 179.39─814.47; $\bar{x}\%$ = 32.45, 95% CI19.60─45.31 and $\bar{X}$ = 1521.14), then in the United States (n = 6; $\bar{x}$ = 570, 95% CI-

**Table 2. Distribution of potential factors across different continents.**

| Ethnicity | Factors | Author [Count] | Mean total populations ($\bar{X}$) | Mean respondents ($\bar{x}$), (95% CI) | Mean respondents ($\bar{x}$ %) (95% CI) |
|---|---|---|---|---|---|
| Asia | Safety | Jain et al., 2021 [28]; Wang K et al., 2021 [30]; Suresh et al., 2021 [31]; Abedin etal.,2021 [32]; Bai et al., 2021 [33]; El-Elimat et al., 2021 [34]; Cai et al., 2021 [35]; Almaghaslah et al., 2021 [36]; Sharun et al., 2020 [39]; Wang J et al., 2020 [42]; Al-Mulla et al., 2021 [43];Rosenta l&Shmueli, 2021 [47]; Jiang et al., 2021 [48]; Burhamah et al., 2021 [53] | 1521.14 | 496.93 (179.39─814.47) | 32.45 (19.60─45.31) |
| | Efficacy | Jain et al., 2021 [28]; Almaghaslah et al., 2021 [36]; Kose et al., 2021 [56] | 1022.67 | 269 (115.70─422.30) | 27.03 (10.35─43.71) |
| | Information sufficiency | Suresh et al., 2021 [31]; Almaghaslah et al., 2021 [36]; Kaplan et al., 2021 [61] | 931.33 | 419.33 (7.93─830.73) | 40 (27.01─52.99) |
| | Trust | Jain et al., 2021 [28];El-Elimat et al., 2021 [34]; Kose et al., 2021 [56]; Padhi& Al-Mohaithef, 2021 [66] | 1574.50 | 275.25 (51.22─499.28) | 16.78 (6.20─27.35) |
| | Side effect | Suresh et al., 2021 [27]; Bai et al., 2021 [33];El-Elimat et al.,2021 [34]; Rosental&Shmueli, 2021 [47]; Jiang et al., 2021 [48];Kose et al., 2021 [56] | 1598.83 | 660.33 (-9.57─1330.23) | 35.78 (16.79─54.77) |
| | Effectiveness | Wang et al., 2021 [30]; Abedin et al., 2021 [32]; El-Elimat et al.,2021 [34]; Almaghaslah et al., 2021 [36]; Sharun et al., 2020 [39]; Wang J et al., 2020 [42]; Al-Mulla et al., 2021 [43] | 1638.71 | 563.57 (180.27─946.87) | 44.84 (25─64.68) |
| | Conspiracy beliefs | Burhamah et al., 2021 [53]; Sallam et al., 2021a [72]; Sallam et al., 2021b [73] | 1598.75 | 273.50 (54.91─492.10) | 18.55 (8.67─28.43) |
| | Social influence | Cai et al., 2021 [35] | 1057 | 332─ | 31.4─ |
| | Vaccine-mandate | Almaghaslah et al., 2021 [36] | 862 | 402 ─ | 46.7─ |
| | Fear & anxiety | Rosental&Shmueli, 2021 [47] | 628 | 112 ─ | 17.8─ |

*(Continued)*

**Table 2.** (Continued)

| Ethnicity | Factors | Author [Count] | Mean total populations ($\bar{X}$) | Mean respondents ($\bar{x}$), (95% CI) | Mean respondents ($\bar{x}$ %) (95% CI) |
|---|---|---|---|---|---|
| Europe | Safety | Soares et al., 2021 [27]; Petravić et al., 2021 [45]; Grochowska et al., 2021 [46]; Holzmann-Littig et al., 2021 [54] | 1826.50 | 423 (-132.07—978.07) | 28.10 (0.96—55.24) |
| | Efficacy | Tavolacci et al., 2021 [55]; Freeman et al., 2021 [57] | 4101.5 | 265 — | 5.45 — |
| | Information sufficiency | Soares et al., 2021 [27]; Sherman et al., 2021 [58] | 1721.5 | 901.5 — | 50.5 — |
| | Trust | Soares et al., 2021 [27]; Petravić et al., 2021 [45]; Grüner&Krüger, 2020 [65]; Freeman et al., 2021 [57];Holzmann-Littig et al., 2021 [54] | 2477.80 | 361.60 (-83.64—806.84) | 24.94 (2.32—47.56) |
| | Side effect | Tavolacci et al., 2021 [55]; Petravić et al., 2021 [45];Szmyd et al., 2021 [68];Sherman et al., 2021 [58]; Freeman et al., 2021 [57]; Holzmann-Littig et al., 2021 [54] | 2799.67 | 300.50 (154.02—446.98) | 16.39 (4.70—28.08) |
| | Effectiveness | Grochowska et al., 2021 [46]; Holzmann-Littig et al., 2021 [54] | 2369.5 | 281.5— | 10.8 — |
| | Conspiracy beliefs | Szmyd et al., 2021 [68] | 1971 | 310— | 15.7 — |
| | Social influence | Holzmann-Littig et al., 2021 [54]; Tavolacci et al., 2021 [55]; Freeman et al., 2021 [57] | 4234.33 | 1207.33 (-965.03—3379.69) | 23.85 (-18.48—66.18) |
| | Political roles | Holzmann-Littig et al., 2021 [54] | 4500 | 206 — | 4.5— |
| | Fear and anxiety | Holzmann-Littig et al., 2021 [54] | 4500 | 302 — | 6.7— |
| The United States | Safety | Lin et al., 2020 [29]; Silva et al., 2021 [37]; Manning et al., 2021 [38]; Palm et al., 2021 [40]; Pogue et al., 2020 [41];Nikolovski et al.,2021 [52] | 2274.67 | 570 (-214.95—1,354.95) | 33.33 (12.68—53.98) |
| | Efficacy | Lin et al., 2020 [29]; Nikolovski et al., 2021 [52] | 5471.5 | 1369 — | 37.55 — |
| | Information sufficiency | Lin et al., 2020 [29]; Silva et al., 2021 [37]; Nikolovski et al., 2021 [52];Lucia et al., 2020 [62] | 2836.75 | 2231.50 (-420.29—4883.29 | 51.43 (14.12—88.74) |
| | Trust | Lucia et al., 2020 [62]; Mascarenhas et al., 2021 [63]; Kelekar et al., 2021 [64] | 276.67 | 94.67 (16.58—172.76) | 34.53 (3.74—65.32) |
| | Side effect | Manning et al., 2021 [38]; Lucia et al., 2020 [62]; Nikolovski et al., 2021 [52] | 2866 | 1025 (-686.87—2736.87) | 36.23 (20.29—52.17) |
| | Effectiveness | Silva et al., 2021 [37]; Palm et al., 2021 [40]; Pogue et al., 2020 [41];Nikolovski et al., 2021 [52]; Lucia et al., 2020 [62];Reiter et al., 2021 [70] | 1875.17 | 173.17 (-7.92—354.26) | 16.68 (8.47—24.89) |
| | Conspiracy beliefs | Pogue et al., 2020 [41] | 316 | 3 — | 1— |
| | Social influence | Lin et al., 2020 [29]; Mascarenhas et al., 2021 [63] | 1894.5 | 1540.5 — | 74.85— |
| | Political roles | Palm et al., 2021 [40]; Reiter et al., 2021 [70] | 1564.5 | 405.5 — | 23.25— |
| | Vaccine-mandate | Silva et al., 2021 [37]; Lucia et al., 2020 [62]; Mascarenhas et al., 2021 [63] | 217.33 | 124.67 (62—187.34) | 52.70 (27.43—77.97) |
| | Fear and anxiety | Nikolovski et al., 2021 [52] | 7402 | 126— | 1.7— |
| Africa | Safety | Kanyike et al., 2021 [44]; Mudenda et al., 2021 [49]; Faezi et al., 2021 [51] | 935.33 | 581 (32.01—1129.99) | 64.73 (58.36—71.10) |
| | Efficacy | Kanyike et al., 2021 [44] | 600 | 376— | 62.7 — |
| | Information sufficiency | Kanyike et al., 2021 [44] Saied et al., 2021 [59] | 1366.5 | 963— | 67.7— |
| | Trust | Kanyike et al., 2021 [44] | 600 | 376— | 62.7— |
| | Side effect | Kanyike et al., 2021 [44]; Mudenda et al., 2021 [49]; Faezi et al., 2021 [51]; Saied et al., 2021 [59] | 1234.75 | 957.75 (140.55—1774.94) | 74.60 (58.08—91.11) |
| | Effectiveness | Saied et al., 2021 [59]; Mudenda et al., 2021 [49] | 1229.5 | 1010.5— | 51.7 — |

(*Continued*)

**Table 2.** (Continued)

| Ethnicity | Factors | Author [Count] | Mean total populations ($\bar{X}$) | Mean respondents ($\bar{x}$), (95% CI) | Mean respondents ($\bar{x}$ %) (95% CI) |
|---|---|---|---|---|---|
| Multi-ethnic areas | Safety | Lazarus et al., 2021 [50] | 13426 | 560— | 4.1— |
| | Information sufficiency | Riad et al., 2021 [60] | 6639 | 2091— | 31.5 — |
| | Trust | Riad et al., 2021 [60]; Lazarus et al., 2021 [50] | 10032.5 | 2978— | 30.3— |
| | Side effect | Riad et al., 2021 [60] | 6639 | 3369— | 50.7— |
| | Effectiveness | Lazarus et al., 2021 [50] | 13426 | 560— | 4.1— |
| | Conspiracy beliefs | Lazarus et al., 2021 [50] Islam et al., 2021 [71] | 7031.5 | 1002— | 11.75— |
| | Political roles | Riad et al., 2021 [60] | 6639 | 1068— | 16— |
| LMICs | Side effect | Bono et al., 2021 [67]; Arce et al., 2021 [70] | 13055.5 | 5283— | 40.6— |
| | Effectiveness | Bono et al., 2021 [67] | 10183 | 1538— | 15.1— |

214.95—1,354.95; $\bar{x}$% = 33.33, 95% CI 12.68—53.98, $\bar{X}$ = 2274.67), followed by Europe (n = 4; $\bar{x}$ = 423, 95% CI -132.07—978.07; $\bar{x}$% = 28.10, 95% CI0.96—55.24 and $\bar{X}$ = 1826.50), Africa (n = 3; $\bar{x}$ = 581, 95% CI 32.01—1129.99; $\bar{x}$% = 64.73, 95% CI 58.36—71.10, and $\bar{X}$ = 935.33), and multi-ethnic areas (n = 1; $\bar{x}$ = 560; $\bar{x}$% = 4.1 and $\bar{X}$ = 13426). "Side effects" was identified and distributed as a potential factor equally (n = 6) in Asia ($\bar{x}$ = 660.33, 95% CI-9.57—1330.23; $\bar{x}$% = 35.78, 95% CI 16.79—54.77, $\bar{X}$ = 1598.83) and Europe ($\bar{x}$ = 300.50, 95% CI 154.02—446.98; $\bar{x}$% = 16.93, 95% CI 4.70—28.08, $\bar{X}$ = 2799.67), followed by Africa (n = 4; $\bar{x}$ = 957.75, 95% CI 140.55—1774.94; $\bar{x}$% = 74.60, 95% CI 58.08—91.11 and $\bar{X}$ = 1234.75); the United States (n = 3; $\bar{x}$ = 1025, 95% CI -686.87—2736.87; $\bar{x}$% (%) = 36.23, 95% CI 20.29—52.17, $\bar{X}$ = 2866), LMICs (n = 2; $\bar{x}$ = 5283, $\bar{x}$% = 40.6 and $\bar{X}$ = 13055.5) and multi-ethnic regions (n = 1; $\bar{x}$ = 3369, $\bar{x}$% = 50.7 and $\bar{X}$ = 6639). The greatest responses to "effectiveness" were found (n = 7) in the studies in Asian countries ($\bar{x}$ = 563.57, 95% CI180.27—946.87; $\bar{x}$% (%) = 44.84, 95% CI 25—64.68 and $\bar{X}$ = 1638.71), followed by the United States(n = 6; $\bar{x}$ = 173.17, 95% CI-7.92—354.26; $\bar{x}$% = 16.68, 95% CI 8.47—24.89 and $\bar{X}$ = 1875.17), Africa (n = 2; $\bar{x}$ = 1010.5, $\bar{x}$% = 51.7 and $\bar{X}$ = 1229.5) and, LMICs and multi-ethnic areas (n = 1; $\bar{x}$ = 560, $\bar{x}$% = 4.1, $\bar{X}$ = 13426; $\bar{x}$ = 1538, $\bar{x}$% = 15.1, $\bar{X}$ = 10183, respectively). In Europe "trust" (n = 5) was distinguished as a critical predictor ($\bar{x}$ = 361.60, 95% CI-83.64—806.84; $\bar{x}$% = 24.94, 95% CI2.32—47.56 and $\bar{X}$ = 2477.80), whereas in Asian countries trust was recognized in only 4 studies ($\bar{x}$ = 275.25, 95% CI 51.22—499.28; $\bar{x}$% = 16.78, 95% CI 6.20—27.35 and $\bar{X}$ = 1574.50) followed by the United States (n = 3; $\bar{x}$ = 94.67, 95% CI 16.58—172.76, $\bar{x}$% = 34.53, 95% CI = 3.74—65.32 and $\bar{X}$ = 276.67), multi-ethnic areas (n = 2; $\bar{x}$ = 2978, $\bar{x}$% = 30.3, $\bar{X}$ = 10032.5), and Africa (n = 1; $\bar{x}$ = 376, $\bar{x}$% = 62.5, $\bar{X}$ = 600). "Information sufficiency" was an important determinant in reducing COVID-19 vaccine hesitancy, which was identified mostly (n = 4) in articles from the United States ($\bar{x}$ = 2231.5, 95% CI -420.29—4883.29; $\bar{x}$% = 51.53, 95% CI14.12—88.74, $\bar{X}$ = 2836.75) followed by Asia (n = 3; $\bar{x}$ = 419.33, 95% CI 7.93—830.73; $\bar{x}$% = 40, 95% CI 27.01—52.99 and $\bar{X}$ = 931.33), and articles (n = 2) from Europe and Africa ($\bar{x}$ = 901.5, $\bar{x}$% = 50.5, $\bar{X}$ = 1721.5; ($\bar{x}$ = 963, $\bar{x}$% = 67.7, $\bar{X}$ = 1366.5, respectively). The public concern regarding the "efficacy" of the COVID-19 vaccine was predominant (n = 3) in the studies conducted in Asian countries ($\bar{x}$ = 269, 95% CI 115.7—422.30; $\bar{x}$% = 27.03, 95% CI 10.35—43.71 and $\bar{X}$ = 1022.67), followed by an equal (n = 2) distribution in the United States and Europe ($\bar{x}$ = 1367, $\bar{x}$% = 37.55, $\bar{X}$ = 5471.5; ($\bar{x}$ = 265, $\bar{x}$% = 5.45, $\bar{X}$ = 4101.5, respectively). Among the total articles we analyzed, "conspiracy beliefs" was explored as one of the key

predictors, especially in Asian countries (n = 3; x̄ = 273.5, 95% CI 54.91⎯492.10; x̄% (%) = 18.55, 95% CI 8.67⎯28.43 and X̄ = 1598.75), followed by multi-ethnic areas (n = 2; x̄ = 1002, x̄% = 11.75 and X̄ = 7031.5). The opinions provided by friends, family, and social networks (social influence) were mostly valued by individual's when making a rapid vaccination decision in Europe (n = 3; x̄ = 1207.33, 95% CI -665.03⎯3379.69; x̄% = 23.85, 95% CI -18.48⎯66.18 and X̄ = 4234.33), followed by the United States (n = 2; x̄ = 1540.5, x̄% = 74.85 and X̄ = 1894.5). It was observed that, information from political leaders directly affected vaccination decisions, particularly in the United States; hence, "political roles" was mostly identified in the United States (n = 2; x̄ = 405.5, x̄% = 23.25 and X̄ = 1564.5). In the same manner, "vaccine-mandated" was a vital issue that was mostly reported in studies (n = 3) from the United States (x̄ = 124.67, 95% CI62⎯187.34; x̄% = 52.7, 95% CI 27.43⎯77.97 and X̄ = 217.33). The influence of negative emotions such as "fear and anxiety" on the COVID-19 vaccine acceptance intention was found in single articles from Asia, the United States, and Europe as shown below in Table 2.

## Discussion

Public acceptance of a new vaccine is not an old concept; rather, it is a dynamic phenomenon that is regulated sharply by psychological behavior, societal issues, and vaccine-derived factors related to a particular vaccine candidate. Since human psychological behaviors change over space, time, and environment, achieving equitable vaccination rates across all population groups indeed is a challenging issue in light of such multifaceted psychological behavior [74]. The human psychological behaviors related to immunization are almost the same in terms of responses to uptake intention for national vaccination programs and protection from a particular pandemic disease [75]. In this complex behavioral patterns, vaccine hesitancy and low initial vaccine uptake for a particular vaccine or even a vaccination program are serious threats to global health, with several common socio-psychological factors having been reported during the outbreaks of measles and pertussis [76] and for influenza vaccination [77]. Importantly, the introduction and distribution of a new vaccine is an economically costly and time-consuming process, while acceptability of a vaccine is the leading indicator that controls the overall success of vaccination programs [78, 79]. As such, estimating and exploring the common factors of COVID-19 vaccine hesitancy is an effective step in designing an action plan for improving the overall acceptance rate.

In our review, the safety, side effects, and effectiveness were identified as the most common predictors of COVID-19 vaccine acceptance around the world. Perceived vaccine safety and effectiveness were seen as the most common factors associated with vaccine hesitancy in previous vaccination programs, as reported in several scientific studies. A comprehensive review of 2791 studies conducted between 1990 and 2019 revealed that, although vaccine hesitancy largely depends on the disease severity, culture and local context, concerns about vaccine safety are the actual cause of vaccine refusal [80]. Another review of 1187 articles primarily on HPV and flu vaccines concluded that, both side effects and safety concerns were the leading causes of vaccine refusal by the general public and health care workers [81]. In the same manner, Karafillakis and Larson (2017)-, performed a review of 2895 English, French, and Spanish studies from 2004 to 2014 and found that, the greatest vaccine concerns were safety and efficacy issues, among other factors [82]. Along with safety concerns, the perceived efficacy of new vaccines was found to be a critical predictor of vaccine acceptance decisions in a study on H1N1 vaccine promotion to the older adults [79, 83]. All together, these results are consistent with the identified factors associated with COVID-19 vaccine acceptance and hesitancy. Vaccine uptake could also be a decreasing function of current or past incidences of side effects that have appeared with vaccination [84]. Chapman and Coups (1999) reported that, side effects and

effectiveness were the most important factors in influenza vaccine uptake decisions by healthy adults [85].

The extent to which the public trust the vaccine to be safe and effective after administration was the strongest forecaster of COVID-19 vaccine uptake intention. In the same way, a recent past study showed that, vaccine confidence levels regarding the safety and effectiveness were influenced by the level of trust in the vaccine, because trust plays a key role in regulating vaccine hesitancy [86]. Larson et al., (2018) performed a systematic review and reported that trust had the greatest impact on vaccine acceptance in low-and middle-income countries (LMICs) [87]. Similarly, misconceptions and mistrust regarding vaccine efficacy were recognized as the most common reasons to refuse the seasonal influenza vaccine for the health care workers in Ireland [88]. Even in national vaccination programs, trust and information sufficiency are the critical predictors needed for parents to make informed decision regarding their children's HPV vaccination [89]. Individual acceptance of vaccination depends not only on knowledge about the risks and benefits of vaccines, but also religious, cultural, emotional, and social factors which are considered the more complex determinants [90, 91]. Restoring public trust in vaccines and the vaccination process was accepted as a key solution to the above aspects [92]; therefore, the critical role of public trust in COVID-19 vaccination has been prioritized as an important factor in our analysis.

We speculated that, conspiracy beliefs and information sufficiency are other important factors in implementing successful vaccination programs in different countries, and along with a lack of trust regarding vaccine benefits, government policies, health systems, vaccine developers and service providers. Additionally, we speculated that hidden and inadequate health information would accelerate anti-vaccine conspiracy beliefs and rumors [93–95]. Accordingly, information sufficiency and conspiracy beliefs were identified as predictors of COVID-19 vaccine acceptance and hesitancy. To ensure vaccine trust, the communication strategies and vaccine delivery techniques to be applied during vaccination should be transparent, honest, accurate, truthful, multimodal, and frequent, involving partnerships with community and health workers in an inclusive manner. Typical communication methods used for health professionals and health policy makers will not be very effective in reaching marginalized groups in improving confidence levels, as the COVID-19 itself is not a typical scenario. In such atypical settings, remote contact strategies are preferred, with information presented a non-professional manner, following a general style that is easily understandable by the general public and communicating the major issues to be addressed. As a result, a more unique, multidisciplinary, organized approach from reputed public health experts, academicians, scientist, health professionals, and local political leaders is needed. A rapid solution for reducing vaccine hesitancy would be to focus on communicating effectively using evidence-based information, counteracting messages that can misinform the general public. Rzymski et al., (2021) emphasized evidence-based communication strategies are essential when dealing with community members in order to control the COVID-19 vaccine-related misinformation and to ensure large public benefits [96]. On the other hand, Arede et al., (2019) focused on long-term communicative approaches to overcome vaccine hesitancy, involving the promotion of vaccine literature through different communication channels. This strategy can work as a fundamental tool for appropriate communication by enabling critical thinking and access to vaccine related health education and information [97].

The positive social influences on the vaccines intention have examined in past articles on HPV [98]. Friends who had already been vaccinated had significant influence on individual's decision to receive the influenza vaccine [99] and flu vaccine [100, 101]. Perceived vaccine effectiveness and social influence were identified as the core determinants of influenza vaccine uptake intention among healthy adults in the United States [102]. Similarly, social influence

has been recognized as an important predictor of COVID-19 vaccination uptake intention in Europe and the United States.

In our analysis, "political roles" was a factor identified in scientific studies from the United States. The general populations of the United States has become sharply divided regarding all aspects of science surrounding COVID-19,ranging from its origin to the government actions and policies seeking to mitigate the pandemic's impacts [103, 104]. In this regard, a content analysis of electronic and print media coverage surrounding the COVID-19 issue showed that politicians were featured as often as or some times more often than public health experts and scientists regarding the COVID-19 issues in the United States [105]. In addition to the above mentioned key factors, some other important factors such as previous vaccine exposure, cultural history, perceived risk of infection, personal consequence, and regional ethnicity were also considered by the general population when deciding to decline or accept a new H1N1 vaccine, as reported in past studies [106–108].

Mass vaccination programs against COVID-19 have been started worldwide; therefore, identifying the factors associated with vaccine acceptance intention and hesitancy is an important consideration that needs to be addressed. While the vaccination process has started, many people in specific regions have remained confused about whether they should take the vaccine or not. A recent review conducted on 13 countries reported that 60% (95% CI 49%─69%) of the sample population had the intention to be immunized by the COVID-19 vaccine [19]; therefore, if the overwhelming majority of the hesitant population is kept outside of the vaccination process, attempts to offer free vaccinations would not be successful in restricting the COVID-19 contamination rates. Given the potential influential factors associated with COVID-19 vaccination consequences, vaccine policy makers should develop guidelines for COVID-19 vaccination on the basis of priority group identification. To reduce pandemic-induced morbidity and mortality rates, COVID-19 vaccine administration should be mandated for elderly and co-morbid individuals, because these groups are more vulnerable to the corona virus than others and there is a strong association of age and co-morbidities with the mortality rate as shown found in a recent systematic review and meta-analysis [109].

## Limitations

The current study had some limitations; the foremost of which was article sample size. The total number of articles we examined was not highly satisfactory in comparison with other scholarly articles published during the current COVID-19 vaccination era. We wanted to emphasize certain on some selective predictors that potentially impact on COVID-19 vaccine receptivity and refusal intention; however, other relevant and important factors may also lead to vaccine refusal, thereby reducing acceptance intention, including socio-demographic characteristics, employment status, perceived risk exposure, cultural differences, personal and professional consequences, doctor recommendations, and inoculation history [110–112]. Ethnicity is also a predictor, along with socio-demographic differences, of accepting COVID-19 vaccines [113] Religious beliefs and rumors in South Asian countries [114] were not included in this analysis. Finally, most of the research studies included in this review employed a cross-sectional survey type, providing snapshots of the vaccine hesitancy status in each country. These studies applied different sampling strategies, which might lack some of the potential factors that are closely associated with the actual vaccine acceptance rates and hesitancy levels reported in different countries.

## Implications

The practical implications of this systematic review in terms of vaccination policy and future research include the following aspect: (a) This review acts as scientific evidence for initiating

further predictive studies of COVID-19 vaccine acceptance and for examining the association between hesitancy and explanatory variables. This study could be helpful in determining the influential factors in countries in which small scale vaccine-delivery has started or has to start. At a press briefing, the director general of the WHO was worried about mismanagement in vaccine distribution and social inequalities globally, because poor countries are still far away from the adequate access to the required COVID-19 vaccine needed for mass immunization of their population. The WHO director general thereby urged drug makers to supply maximum amounts of COVID- 19 vaccines to poor countries on a priority basis [115]. As a result, there is an opportunity for researchers in these countries to identify the potential factors in advance and implement effective policies on vaccine delivery to accomplish crude vaccine coverage. (b) The upcoming studies will pave the way in identifying the key influential factors of vaccine hesitancy in particular regions, thereby supporting efforts to estimate the effect size of such factors towards the acceptance intention of the general public. (c) This study could motivate health students and health care workers to describe their experiences of the influence of potentially related factors and could encourage them to engage in adequate vaccine health campaigns. Such efforts help in reinforcing the building of sustainable trust levels and accelerating the COVID-19 vaccination progression in marginalized areas. (d) This study shows the way to determine country-specific reasons for vaccine hesitancy in order to develop mitigation strategies that would ensure high and equitable vaccination coverage across LMICs. (e) This study will largely benefit health policy makers and vaccine promoters in different countries to design evidence-based promotional strategies that will enhance public engagement in the COVID-19 vaccine roll-out.

## Conclusions

The reluctance towards and refusal of COVID-19 vaccines is currently a global concern. Large variability in COVID-19 vaccine acceptance rates has been clearly reported all around the world [19, 24]. In this study, we explored and described 11 potential common factors, among which safety, efficacy, side effects, effectiveness, and conspiracy beliefs were identified most frequently from the studies in Asian countries. In Europe, side effects, trust, and social influence were the predominant influences on decision to receive a COVID-19 vaccine, while information sufficiency, political roles, and vaccine-mandates in the United States. Although the prevailing vaccine resistance factors may vary widely depending on the geographic location, it is clear from the results reported in this review that global COVID-19 vaccine acceptance is dependent on several common psychological, societal, and vaccine related factors. Investigating the key influential factors of COVID-19 vaccine hesitancy is a fundamental task that must be undertaken to ensure an effective COVID-19 immunization plan worldwide.

A major challenge to the successful implementation of COVID-19 vaccination programs is the unpredictable nature of the pandemic. The adequate manufacture of vaccines and proper distribution, vaccine safety confirmation, uncertainty regarding long-term efficacy, and the acquisition of optimal immunity are other challenges that must be overcome. Public trust in health systems and in the vaccine information provided by government agencies regarding vaccine safety, efficacy, and side effects as well as the communicative roles of the media and public health experts will also be essential in improving vaccine confidence among rural and disadvantaged groups in low-income countries. Useful communication channels and public trust in vaccinations will remove anti-vaccine beliefs, fear, anxiety, and rumors, thereby enabling rapid vaccine uptake. Regular, follow-up and timely communication during the pandemic could be important drivers of vaccine confidence and in maintaining peak trust among population groups. Effective messages clarifying the safety, effectiveness, and side effects of

COVID-19 vaccines will increase public trust and promote vaccine confidence among less-educated and doubtful individuals in rural places. In summary, the policy makers should focus on the effects of psychological, societal, and vaccine-related factors, which may be associated with the uptake intention and lead to vaccine hesitancy in a particular territory. To ensure the prompt achievement of herd immunity, the scientific community and health authorities should pay attention to and validate potential common and individual factors, and the potency with which they may influence COVID-19 vaccine acceptance and hesitancy in a given geographical location.

## Supporting information

**S1 Checklist. PRISMA 2020 checklist.**
(DOCX)

**S1 Dataset.**
(DOCX)

## Acknowledgments

All authors greatly acknowledge the graduate students of Jashore University of Science and Technology, who were sincerely assisted in literature search process.

## Author Contributions

**Conceptualization:** Debendra Nath Roy, Md. Shah Azam.

**Data curation:** Debendra Nath Roy, Mohitosh Biswas, Ekramul Islam, Md. Shah Azam.

**Formal analysis:** Debendra Nath Roy, Mohitosh Biswas, Ekramul Islam, Md. Shah Azam.

**Investigation:** Debendra Nath Roy, Mohitosh Biswas, Ekramul Islam, Md. Shah Azam.

**Methodology:** Debendra Nath Roy, Mohitosh Biswas.

**Project administration:** Ekramul Islam, Md. Shah Azam.

**Resources:** Debendra Nath Roy.

**Software:** Debendra Nath Roy.

**Supervision:** Ekramul Islam, Md. Shah Azam.

**Validation:** Mohitosh Biswas, Ekramul Islam, Md. Shah Azam.

**Visualization:** Debendra Nath Roy, Mohitosh Biswas, Ekramul Islam.

**Writing – original draft:** Debendra Nath Roy.

**Writing – review & editing:** Mohitosh Biswas, Md. Shah Azam.

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
