## [Decision Letter · Decision Letter 0]

12 Oct 2021

PONE-D-21-26646Potential factors influencing COVID-19 vaccine acceptance and hesitancy: A systematic reviewPLOS ONE

Dear Dr. Azam,

Thank you for submitting your manuscript to PLOS ONE. After careful consideration, we feel that it has merit but does not fully meet PLOS ONE’s publication criteria as it currently stands. Therefore, we invite you to submit a revised version of the manuscript that addresses the points raised during the review process.

The paper addresses an interesting topic in the now-a-days context. I think that the authors should try to improve the paper based on the comments made by the reviewers.

We look forward to receiving your revised manuscript.

Kind regards,

Camelia Delcea

Academic Editor

PLOS ONE

Journal Requirements:

“No external fund was available.”

Reviewers' comments:

Reviewer's Responses to Questions

**Comments to the Author**

1. Is the manuscript technically sound, and do the data support the conclusions?

Reviewer #1: Partly

Reviewer #2: Yes

2. Has the statistical analysis been performed appropriately and rigorously? 

Reviewer #1: N/A

Reviewer #2: Yes

3. Have the authors made all data underlying the findings in their manuscript fully available?

Reviewer #1: Yes

Reviewer #2: Yes

4. Is the manuscript presented in an intelligible fashion and written in standard English?

Reviewer #1: No

Reviewer #2: Yes

5. Review Comments to the Author

Reviewer #1: The article addresses an important and timely question, seeking to synthesize the available evidence related to factors influencing COVID-19 vaccine acceptance and hesitancy. Although the need for such integrative work is conspicuous, I believe that this work in its current shape does not fit the standards of PLOS ONE, warranting a major revision.

My chief concern is with the subpar quality of writing. The entire text is rife with awkward wordings and unclear formulations, which oftentimes render it difficult to follow the narrative. In this genre of academic text – integrative reviews – clarity of presentation and clear-cut language are of utmost importance. I suggest that the team of authors enlist outside help, perhaps a professional copy-editing service, to give the text a thorough makeover before a decision can be made on the substantive merit of the work.

Methodologically, I am not convinced that the search protocol described in lines 167-174 of the text would produce a comprehensive body of texts required to answer the questions that the study seeks to answer. It seems redundant to rely on a string of search terms that include “COVID-19 vaccine hesitancy” and additional terms that, in some cases, can lead to getting at preordained factors (e.g., “COVID-19 vaccine hesitancy and religiosity/conspiracy beliefs”), rather than helping the researchers to discover the full range of factors discussed in the literature. Furthermore, the choice to include terms that are this specific should be derived from theory and explained in the protocol development section.

Reviewer #2: In the actual context, the paper presents a theoretical and practical importance. It is quite interesting the analysis of the eleven factors which determine the acceptence or hesitancy towards the Covid-19 vaccines. However, the paper was hard to be read because the paragraphs were numbered in the beggining.

6. PLOS authors have the option to publish the peer review history of their article (what does this mean?). If published, this will include your full peer review and any attached files.

Reviewer #1: No

Reviewer #2: No

---

## [Author Response · Author response to Decision Letter 0]

10 Feb 2022

A separate file has been attached titled 'Response to reviewers'

---

## [Decision Letter · Decision Letter 1]

3 Mar 2022

Potential factors influencing COVID-19 vaccine acceptance and hesitancy: A systematic review

PONE-D-21-26646R1

Dear Dr. Azam,

We’re pleased to inform you that your manuscript has been judged scientifically suitable for publication and will be formally accepted for publication once it meets all outstanding technical requirements.

Kind regards,

Camelia Delcea

Academic Editor

PLOS ONE

Additional Editor Comments (optional):

Reviewers' comments:

Reviewer's Responses to Questions

**Comments to the Author**

1. If the authors have adequately addressed your comments raised in a previous round of review and you feel that this manuscript is now acceptable for publication, you may indicate that here to bypass the “Comments to the Author” section, enter your conflict of interest statement in the “Confidential to Editor” section, and submit your "Accept" recommendation.

Reviewer #2: All comments have been addressed

2. Is the manuscript technically sound, and do the data support the conclusions?

Reviewer #2: Yes

3. Has the statistical analysis been performed appropriately and rigorously? 

Reviewer #2: Yes

4. Have the authors made all data underlying the findings in their manuscript fully available?

Reviewer #2: Yes

5. Is the manuscript presented in an intelligible fashion and written in standard English?

Reviewer #2: Yes

6. Review Comments to the Author

Reviewer #2: The paper provides updated datas and important information regarding a subject that has been influencing our lives for the past two years.

I keep my opininion that the paper was well written and had a good background documentation.

7. PLOS authors have the option to publish the peer review history of their article (what does this mean?). If published, this will include your full peer review and any attached files.

Reviewer #2: No

---

## [Editor Report · Acceptance letter]

14 Mar 2022

PONE-D-21-26646R1 

*Potential factors influencing COVID-19 vaccine acceptance and hesitancy: A systematic review*  

Dear Dr. Azam:

I'm pleased to inform you that your manuscript has been deemed suitable for publication in PLOS ONE. Congratulations! Your manuscript is now with our production department. 

Kind regards, 

on behalf of

Dr. Camelia Delcea 

Academic Editor

PLOS ONE